# Towards Culture-Aware Smart and Sustainable Cities: Integrating Historical Sources in Spatial Information Infrastructures

Bénédicte Bucher [1,*], Carola Hein [2], Dorit Raines [3] and Valérie Gouet Brunet [1]

1   LASTIG Lab, Université Gustave Eiffel, IGN-ENSG, 94 160 Saint Mande, France; valerie.gouet@ign.fr
2   Faculty of Architecture and the Built Environment, Delft University of Technology, P.O. Box 5, 2600 AA Delft, The Netherlands; c.m.hein@tudelft.nl
3   Department of Humanistic Studies, Università Ca' Foscari Venezia, Dorsoduro 3246, 30123 Venice, Italy; raines@unive.it
*   Correspondence: benedicte.bucher@ign.fr

**Abstract:** This article addresses the integration of cultural perspectives in the smart city discourse and in the implementation of the UN Agenda 2030; it does so specifically with respect to land patterns and land use. We hope to increase the ability of relevant stakeholders, including scientific communities working in that field, to handle the complexity of the current urban challenges. Culture is understood here in the broadest sense of the word, including the values and conceptualizations of the world, and the modes of technological creation and control of the environment. This concept of culture varies among stakeholders, depending, in particular, on their activities, on the place they live in, and also depending on their scientific background. We propose to complement existing targets that are explicitly related to culture in the UN and UNESCO agendas for 2030, and introduce a target of culture awareness for city information infrastructures. We show that, in the specific case of land patterns and land use, these new targets can be approached with historical data. Our analysis of the related core functionalities is based on interviews with practitioners, draws on insights from the humanities, and takes into account the readiness of the existing technologies.

**Keywords:** smart cities; cultural; land use; historical data; multimodality; synergy; spatial information infrastructures; linked data

## 1. Introduction

Making cities more inclusive and sustainable is a widely acknowledged necessity. In 2015, all the United Nations member states adopted the UN Agenda 2030, which provides guidelines for achieving the global change that is necessary to eradicate poverty in all its forms and to protect the planet [1]. Among the seventeen Sustainable Development Goals (SDGs) that are identified in this agenda, one goal, SDG11, aims to "make cities and human settlements inclusive, safe, resilient and sustainable". It is associated with ten targets that cover different aspects of this ambitious goal and fifteen indicators of progress towards those targets. Two important aspects of this agenda should be underlined here. The first is that it encourages partnerships between stakeholders, as a necessary solution to the crisis that is faced by our planet and societies. The second is that goals, targets and indicators are not intended to partition our world into subdomains, but rather to provide a means to survey and understand a complex reality through statistics. As we will explain, the current smart cities discourse has evolved from its origins in the 1990s, to address how cities must transform to face the contemporary challenges of natural resource declines, climate change, and digital transformation.

Culture is an important asset for this transition, but is currently under-rated. UNESCO outlined a culture 2030 Agenda, complementary to the UN Agenda 2030, to highlight and increase the contribution of culture to the SDGs. Culture 2030 emphasizes cultural heritage, creative industry, creativity and innovation, local communities, local products and

materials, as well as cultural diversity [2]. We argue that a more comprehensive cultural perspective, in the widest sense, including technological and social aspects, is needed in the smart cities discussions, in research and in practice, to foster partnerships between relevant scientific communities and the general and dynamic field of smart cities, and to engage local stakeholders in the UN Agenda 2030. It is important to understand the cultural contexts and specific visions of the world, modes of technological creation, and efforts to control the natural environment developed by individuals and communities [3], before evaluating and combining different technological approaches to the smart and sustainable city domain, and developing truly smart and sustainable solutions that are embraced by citizens.

Our first hypothesis is that it is possible to improve the adoption and prioritization of this somewhat general objective, by defining a shareable and concrete target that is related to it. We propose that this new target can be defined for the digital representations of the city through data, which is implicit and transversal to all the SDGs, as well as to the smart city approaches. This new target of culture awareness for smart cities information infrastructures complements the existing targets in the UN Agenda 2030 and culture 2030.

We propose studying land use as a way to demonstrate opportunities for such a culture-based approach. Land use, which is the way that we occupy space, is key to the future of cities. It is related to local stakes, such as greening a city district, and to more global stakes, such as diminishing the greenhouse gas emissions of our planet, to every citizen's experience, and to global negotiations, and is a domain where adopting cultural perspectives is especially needed. The following is just one example: in many countries, the decision to allow suburbanization in the post-war period and the promotion of single-family homes both required and was enabled by extensive construction of infrastructure, including highways.

Concrete pathways are needed to achieve culture-aware information infrastructure for cities. Our second hypothesis is that integrating historical data on land use with information infrastructure for smart cities is a concrete pathway to account for the palimpsestic process of land use and facilitate the understanding of how values of the past are written into contemporary spaces, potentially determining the ways of life and forms of design in the future. The LandUseWheel summarizes the required functionalities to achieve culture awareness of city land use information infrastructures, based on historical data. Its design is based on interviews with practitioners, and it has also been influenced by insights from the humanities and from geographical information science. Three key technologies to support the LandUseWheel are identified and their technology-readiness is briefly analyzed, based on the different projects of the authors.

In the next section, following a brief description of the context of this research and of the field of smart cities, we define our proposed target for information infrastructures culture awareness and explain the specific focus on land use. The section that follows presents the first results on the integration of historical data in city information infrastructures, to progress towards the target of the LandUseWheel.

## 2. Introducing a Target of Culture Awareness for Smart Cities

### 2.1. Context of the Research

Contributing to inserting long-time perspectives and cultural dimensions into data-driven smart city development requires an awareness of the ways in which historic practices have determined the spaces that we live in and the institutions that support them. It is also necessary to add more value-based approaches to the technology-driven smart city. This proposal stems from various projects, where the authors of this paper developed distinct contributions from their own specific fields, as follows: digital humanities, spatial data infrastructures, computer vision and multimedia, and history. The proposals that are related to metadata and spatial data infrastructures were developed during seminars of the European organizations EuroGeographics and EuroSDR [4,5], as well as during the URCLIM project on urban climate services, funded by the ERA-NET for climate ser-

vices [6]. The concept of the WaterWheel was developed at the Digital Humanities Group at Delft University of Technology [7]. It is closely related to the works of the Leiden-Delft-Erasmus PortCityFutures Center, for example, which aims to connect space, society, and culture, in port city territories. Content-based retrieval methods were developed during the French projects ALEGORIA and Archival City. ALEGORIA develops multimodal and content-based indexing and visualization tools, to facilitate the promotion of iconographic institutional funds collections, describing the French territory in various periods, from the interwar period to our days [8]. Archival City relies on smart and dedicated tools for accessing, viewing, and using the city archives of the following specific places: Greater Paris, Algiers, Bologna, Chiang-Mai, Jerusalem, and Quito [9].

Between 2018 and 2019, all the authors contributed to the coordinated support action (CSA) Time Machine, funded by the H2020 FET Flagship call. The CSA is a one-year project that delivers ten-year road maps for Europe, which correspond to candidate Flagship projects. The CSA Time Machine focused on the valorization of archives in the big data of the past, to bring a long-term perspective within the current data-centered applications and in artificial intelligence. Although the Horizon framework program did not take over FET Flagships, the participants still worked on these roadmaps. All the co-authors of this paper participated in the definition of a roadmap, to valorize TM technology in different application domains, including smart cities, land use and territorial policies [10]. The methodology of the work was to conduct interviews in different European countries and to consolidate findings from these interviews during working sessions. It was an intensive experience of multidisciplinary dialogue.

### 2.2. The Field of Smart Cities in Search for More Synergy and Adoption

The concept of a smart city was introduced in the 1990s, by the industry of information and communication technologies (ICT), to designate the use of these technologies to improve city functions. It became a label for cities to promote their advanced usage of ICT. This smart city vision can be summarized around three general usages of ICT. First, a smart city provides urban decision makers with an accurate representation of the city, thanks to sensors and real-time observations that are gathered through the internet. Second, it supports decisions, thanks to algorithms that process all the collected data related to the city, and can possibly simulate the behavior and interface of urban entities of the real world, in so-called digital twins. Third, it simplifies and improves the interface between citizens and the government, typically through on-line services.

Other approaches to improve city functions prioritized the city's capacity to be resilient, sustainable, and more inclusive, inspired by European cities that have more of a tradition of encouraging social inclusion [11]. The projects encouraged citizens to adopt sustainable actions and frugal solutions, as well as to develop solidarity in the face of crises [12,13]. Wisdom became a key characteristic of a city that used natural resources rationally, pursued smart growth policies, and controlled urban sprawl [14–16].

With the publication of the UN Agenda 2030, in which indicators are established at the level of countries, the ICT-driven approaches for smart cities positioned themselves towards improving the role of cities, to achieve the SDGs within the international forum coordinated by the International Telecommunication Union (ITU), called the United 4 Smart Sustainable Cities (U4SSC). The purpose of the U4SSC is to support cities in their contribution to all the SDGs. The ITU defined the smart and sustainable city as an "innovative city that uses information and communication technologies (ICTs) and other means to improve quality of life, efficiency of urban operation and services, and competitiveness, while ensuring that it meets the needs of present and future generations with respect to economic, social, environmental as well as cultural aspects" [17]. The U4SSC introduces key performance indicators (KPI) for cities. It identifies frontier technologies, namely, artificial intelligence, Internet of Things (IoT), digital twins, unmanned aerial vehicles/drones, wearable technology, and virtual/augmented reality, which can serve the smart and sustainable

city. The U4SSC also proposes guidelines based on simple technologies, to adapt to every city capacity.

In 2018, the technical committee 268 of the International Organization for Standardization "Sustainable Cities and Communities" proposed a new definition of a smart city, as a "city that increases the pace at which it provides social, economic and environmental sustainability outcomes and responds to challenges such as climate change, rapid population growth, and political and economic instability". The definition explains the method of the smart and sustainable city, stipulating that it "fundamentally [improves] how it engages society, applies collaborative leadership methods, works across disciplines and city systems, and uses data, information and modern technologies to deliver better services and quality of life to those in the city (residents, businesses, visitors), now and for the foreseeable future, without unfair disadvantage of others or degradation of the natural environment" [18]. The ISO TC 268 investigated different fields of standardization for the smart and sustainable city, for example, a list of standard indicators of contribution to the SDGs, and toolkits for good practices and operations. It noted that each city must define its own smart city strategy, through a discussion and debate between interested parties, and that a standardization approach should not propose "a one-size-fits-all model for the future of cities", but rather focus on "enabling processes by which innovative use of technology and data, coupled with organizational change, can help each city deliver its own specific vision for a sustainable future in more efficient, effective and agile ways" [19].

Another evolution of the smart city is the application of the concept of a digital twin to the perimeter of an urban area, possibly a city. A digital twin is a digital model of a physical asset that can be updated in real time, based on measures from in situ sensors, located on the asset. It allows the exploration of a large panel of multimodal data attached to the asset. It makes it possible to simulate an operation on the twin and evaluate its impact, before running the corresponding operation on the real asset. As observed by the UN GGIM report, "early examples of digital representations of city infrastructure [in digital twins] have enabled municipalities to monitor and simulate scenarios related to climate change and flooding events while mitigating risks and increasing infrastructure resilience" [20].

Beyond digital twins, it is necessary to develop new forms of intelligence for the smart and sustainable city, to empower collective intelligence, and consolidate public and private interests. The collaborations between ICT specialists and artists aim at encouraging creativity [21]. Specific intelligence and simulation strategies are needed to explore the domain of possible futures, since cities are too complex to be predictable in the medium and long term [22]. Laurini and colleagues [23] highlight the necessity for local authorities to develop territorial intelligence, as companies develop business intelligence. The authors insist on the key role of geographic knowledge bases that should be legible and editable by every stakeholder, and that can also be processed by artificial intelligence, as much as possible. They argue that such territorial intelligence can, for example, reduce the pollution of public debates by the phenomenon of NIMBY (not in my backyard!), where private interests are confused with public stakes.

To date, the scientific and technological domain of the smart city does not refer to a consistent multidisciplinary community sharing one vision, but rather to a domain in search of more dialogue and synergies. There are different approaches that do not always acknowledge one another, and even sometimes reject one another, as was already the case in 2014 [24]. Different labels have been introduced, such as smart, inclusive, sustainable, creative, resilient, or wise, because authors saw possible conflicts between their approaches. Indeed, local inconsistencies exist. A very simple example of conflict is that advancing the actual usage of ICT, to improve the existing functions of the city and the existing public services, involves a short-term perspective that can have a negative impact on inclusiveness and on a city's ecological footprint; the development of digital public services leads to closing physical counters and widening the numeric fraction, and it also may lead to the consumption of more energy. Nevertheless, since 2016, the different approaches are slowly converging. This is partly related to the UN proposal of a 2030 Agenda, and the

identification of universal goals; data are necessary to find a way to reach those goals and to monitor the progress.

This varied and dynamic domain still needs to be adopted by more citizens and decision makers. During the 2019 Digital Cities Challenge Conference, the Siemens company pointed out the very large number of prototypes and concepts, but the lack of a large-scale rollout, one reason being that the current smart cities strategies are driven by the private sector instead of a political agenda. The adoption by local administrations is evolving positively. The French company Data Publica estimates that, in 2017, only about thirty municipalities had engaged in "smart territory" projects, and that in 2020, nearly two hundred French municipalities had integrated digital monitoring of public policies in their calls for tenders. In 2021, a survey taken by this actor, on the hundred largest French municipalities, revealed that 95% have expressed a specific mission related to digital stakes in the local administration, and twenty-three of them explicitly mentioned the smart city or smart territory in the mission title. The same survey was taken for the hundred largest municipalities clusters, which are formal gatherings of municipalities whose population density is lower than the national average; 86% municipalities had expressed such a mission, and twelve titles explicitly mentioned smart city or smart territory. This evolution may be explained by the will of local administrations to preserve city sovereignty and, thanks to smart city platforms, prevent private industries from becoming the key decision makers of the city. It may also be related to a change of strategy in an industry that concentrates primarily on infrastructure, leaving the field of data to the public sector.

To summarize this analysis of the smart city field, there is a growing recognition among scientific communities, but also in the public and private sectors, that a city needs to develop improved functions, using opportunities offered by ICT, and also that a city needs shared objectives and values. Besides, every city should be capable of being smart in its own specific way, with its inhabitants and related societies. To achieve this vision, more synergies are still required between different city stakeholders, and in science between the different communities contributing to smart city solutions, and in the transfer of knowledge from science to society.

*2.3. A Proposed Target of Cultural Awareness for City Information Infrastructures*

In this situation, culture is an important asset to consider for the smart city, in practice and in science. More than one hundred years ago, Tylor [25] defined culture as "an umbrella term which encompasses the social behaviour and norms found in human societies, as well as the knowledge, beliefs, arts, laws, customs, capabilities, and habits of the individuals in these groups". In their 1982 report on cultural policies, UNESCO experts underlined the importance of not understanding culture "in the restricted sense of belles-lettres, the fine art, literature and philosophy, but as the distinctive and specific features and the ways of thinking and organizing their lives of every individual and every community" [3]. Moreover, they argued that culture is a given society's "vision of the world, as much as their mode of scientific and technological creation and control of their natural environment".

UNESCO claims that culture is yet under-evaluated in the UN Agenda 2030, where it is understood as a world's cultural and natural heritage to protect (target 11.4 "strengthen efforts to protect and safeguard the world's cultural and natural heritage"), as a specific sector of activity that encompasses art, creative industry, museums, and also in the widest sense of local habits, knowledge, and products, such as in the target 8.2, which directs participants to "devise and implement policies to promote sustainable tourism that creates jobs and promotes local culture and products". UNESCO proposes a complementary agenda, the Agenda Culture 2030, to evidence, promote, and improve the role of culture, to achieve the SDGs (2), through indicators such as "evidence of management plan(s)/policies/measures to support traditional forms of land ownership and land management elaborated in the last 5 years" and "evidence of integrating cultural factors, including knowledge, traditions and practices of all people and communities, into local strategies on environmental sustainability". The Culture 2030 indicators are intended to be used by nations and also, some of

them, by cities. A specific focus is the design of the methodology, to collect and analyze qualitative and quantitative data, and a methodology to provide evidence of the impacts. UNESCO emphasizes involving local stakeholders in iteratively refining the proposed indicators in a so-called aspirational, rather than normative, reporting.

To meet the challenges of the domain, presented in Section 2.1, a more comprehensive cultural perspective is needed for the smart city discourse, to improve the dialogue and collaboration among actors and communities across different cultures. Our approach is to consider the very abstraction and digitization of the city, which is implicit and transversal to all the SDGs as well as to the smart city approaches. That abstraction should reflect the cultural specificities of the reality at stake, both geographical and societal. It should also adapt to the different backgrounds and values of the stakeholders who participate in the debate or decision. For example, the representation of public space in participatory platforms should be adapted to different participants, whatever their activities, possibly newly arrived migrants who may value places in different ways. This abstraction should also adapt to the scientific and technological backgrounds of different communities, such as ICT specialists, AI, urbanists, sociologists, and artists, encouraging them to acknowledge each other's hypotheses and methods.

Information infrastructures provide ways to translate this general objective into a shared concrete target. Hanseth [26] identifies "information infrastructures" as a notion that is better adapted to our information societies than "information systems", because an information system is isolated and dedicated to a well-identified set of applications, whereas most IT solutions now exchange information with one another. Information technology for cities has evolved over time, in particular in the last thirty years with smart city projects, and will continue to evolve. Hanseth [26] observes that information infrastructures have the following internal dynamic: "all actors have limited influence over their infrastructures [..]. They can at best try cultivating what appears [to be] a living organism". We propose a new target for city information infrastructures, to encourage their further evolution towards supporting cultural perspectives.

> *The target "culture awareness of a city information infrastructure" is defined as follows: the capacity of a city information infrastructure to be used to associate an object of interest, within a city, to information accounting for the related societies, communities and cultural context, and to evaluate the specificities and distinctiveness of this object.*

This target is intended to be understood and adopted by communities, such as those related to the humanities, and also information specialists whose collaboration is required in achieving the goal. While culture is a particularly polysemic word, UNESCO [3] recalls that culture is not only what is often designated as "the cultural sector". The definition does not list everything that is embraced by culture, but extends to other aspects of the UNESCO definition of culture; the accent is put on the capacity to associate an object of interest with related societies and communities, and, secondly, on the capacity to evaluate its specificities and distinctiveness. The object of interest can be a real-world entity, for example, a building. In that case, the information infrastructure should help any user associate it with the related societies and communities—e.g., the designers, builders, and owners—and to evaluate the building's specificities and distinctiveness, for example, in terms of shape, materials, or famous occupants. The object of interest can be an abstract feature, for example a model of a specific building or a taxonomy. In the latter case, the information infrastructure should help any user associate it with the related societies and communities—typically the provenance, and usage of these abstractions—and evaluate its specificities and distinctiveness, such as a specific scope, license, or documentation.

### 2.4. Application to Land Use

The importance of connecting culture to smart city developments can be illustrated through the study of land use. Land use is important in how a city transforms and is, in essence, a cultural object. Which spatial function is located where is a cultural decision and

reflects societal values. This section explains the relevance of culture awareness for city land use information infrastructure.

Culture is relevant to a variety of values and practices associated with land. Stakeholders may have a different understanding of a city's current land use and visions for its future. There has never been unanimity regarding land use. Let us consider the city of Venice. Venice is constantly threatened by flooding. It has been observed that since the 1880s, the sea level has been constantly rising, and flooding is not only increasingly recurrent throughout the year, but the flooding level is increasing in height. A recently completed moving dam, called MOSE, inspired throughout the two decades of its construction energetic discussions and controversy at all levels of society. Naturalists pointed to the danger of closing the three entrances of the Adriatic Sea to the Venetian lagoon too frequently. They claimed that it will undermine the lagoon's biodiversity, by changing the water currents and reducing the oxygen in the water. Fishermen were angry that they could not move freely to the Adriatic Sea, and were constrained to wait long hours until the dam opened. Cruise companies opposed it because they could not determine exactly when their ships would arrive at the Venetian port.

Residents, shopkeepers and, above all, cultural institutions that maintain Venetian monuments (especially in the area of St. Mark's Place), expressed concerns regarding the decision to use the dam only when the sea level during flooding reaches 130 cm (which means that around 60% of the city would be flooded), and they demanded that the level be reduced to at least 110 cm. Raising the dam each time has an economic cost, but not raising it has implications for the preservation of monuments, the city economy, and lagoon biodiversity. The question that lies at the heart of the debate over the moving dam is essentially a cultural one and, put bluntly, it is the following: what is more important—saving the city or its lagoon? [27]. Looking back to the times of the Republic of Venice, one discovers that similar problems existed, but the Venetians' attitude was different; they considered the lagoon and the city as one unity, indivisible, where one influences and, at the same time, sustains the other. Yet, assessing culture's role in a city's land use has been shadowed by administrative and political processes until now, which are believed to be the main engine behind a city's land use. Values and practices that are associated with land use can also differ between locations, for example, between different port cities with different institutional histories.

Decisions about what land to assign to port or city functions, and what environmental or health impacts to accept from (port) industries, comes down to a question of values and culture, and the answers are very different, even in European cities, such as Hamburg, Rotterdam, and London, and certainly in other parts of the world, where different political, economic, and cultural systems prevail [28]. These differences have an important impact when debating at the regional, national, European, and possibly international level. In Europe, every city depends, in some way, on decisions that can be taken at the European level, to conform with Europe's ambition to become a green continent. From that perspective, it is necessary to apply the same regulation to lands with comparable characteristics, to ensure that regulation is fair, which is accomplished by defining shared definitions and classifications of land cover and land use. Europe also aims to become a key player concerning solutions to the challenges faced by cities, such as the environmental footprint, social segregation, mobility, aging, and urban health, thanks to its extensive experience in policy design and thanks to the interaction between the European Union institutions and European cities [29]. In such debates, it is important to distinguish the cultural specificities of different cities, to support bottom-up contributions, instead of imposing standard indicators. As the experts of the Joint Programming Initiative for Cultural Heritage put it, "Europe is a multi-faceted society and its cultural richness is based on the preservation of this diversity including minorities. It is necessary to understand and implement solutions to foster the role of cultural heritage as a factor of cohesion in such a diversified community" [30].

Different cultural perspectives of land use can also relate to different scientific and technological approaches to monitoring, studying, and planning land use, particularly through information infrastructures. A widely adopted approach is to rely on land use and land cover as two complementary notions, to understand the impact of human activity and, recently, of climate change on the surface of the Earth, as well as to transform human activity through regulation. In the European directive on a spatial information infrastructure for Europe, land use is defined as the "territory characterized according to its current and future planned functional or socio-economic purpose (e.g., residential, industrial, commercial, agricultural, forestry, recreational)", and land cover is defined as the "physical and biological cover of the earth's surface including artificial surfaces, agricultural areas, forests, (semi-) natural areas, wetlands, water bodies" [31]. The provision of land cover and land use data can be undertaken at different political levels—local, regional, national, European, international. The following multidisciplinary approach is required to specify, produce, and valorize these data: geographical information science to define and produce land cover and land use data, and natural science and social science to investigate the cause and consequences of land use and land cover changes across a range of spatial and temporal scale [32]. In particular, land surveys across different spatial and temporal scales and extents require cross-level interoperability. The dialogue and cooperation between communities promoting different technologies for monitoring and planning land use could benefit from more culture awareness, in the sense of the capacity of a stakeholder to assess the specific context in which a given technology that is related to land use has been designed and valorized. There is a need for more synergy between different technological perspectives on monitoring and describing land use and land cover in general in Europe, and between INSPIRE (which investigates the reuse of national data) and Copernicus (which investigates the usage of pan-European data acquisition and automated processing).

Focusing on land use, we refine the target that was introduced in the previous section, as follows:

> *The target of the "culture awareness of a city information infrastructure with respect to land use" is defined as follows the capacity of a city information infrastructure to be used to associate a land use of interest to information accounting for the related societies, communities and cultural context, including technologies to create the land use model itself, and to evaluate its specificities and distinctiveness, in comparison with other land uses, other lands and other scales.*

## 3. Cultivating Culture Awareness for Smart Cities' Land Use Information Infrastructures: The LandUseWheel

This section presents a pragmatic approach to making progress towards the target that was defined in Section 2.3, of improving the culture awareness of a city's information infrastructure, with respect to land use.

### 3.1. Grounding Culture Awareness on Historical Data and a Junction between Past, Present and Future

Our approach relies on rendering the cultural specificities of land use in relation to the following three segments: the past, the present, and the future.

Land use patterns have emerged over time, and are based on earlier demarcations. As shown by the historical port city borders in Hamburg, Rotterdam, and Koper, for example, these delimitations can change over time, by location and by function [33]. Yet, the earlier functions remain embedded in the soil, in the infrastructure, and in the buildings. Society builds its knowledge concerning land use in a process that begins in the past and projects into the future. Developing sustainable patterns and cities requires a clear positioning in time, and value-based rethinking of future planning, to make interventions sustainable.

Until some decades ago, the *longue durée* approach was valued, and it permitted not only historians, but also practitioners, to understand patterns and build possible scenarios. Today, when *short-termism* prevails, historical data tend to be excluded from the equation of

city planning [34], and yet, cities and landscapes are the outcome of hundreds of years of interaction between space and institutions; the ways in which politicians and planners think and act are the outcome of historical decisions. For example, in Venice, the morphology of the lagoon has been considerably changed by natural processes, and the present island of Venice was itself created by humans from hundreds of tiny islands. From the Medieval times until today, city authorities relied on land reclamation. Other examples, where information from the past is relevant, include the cases of soil pollution, where practices from the past (such as leatherworking, oil refining, industry, vegetable crops) affect today's soil. Land ownership, land use, land prices, zoning, laws and policy decisions are more generally the result of palimpsestic processes, integrating local, national, and even global forces. The decisions made in the past determine the location of functions, of densities, of pollution, or of land prices. There are patterns and path dependencies—to use a term from the political sciences—of the past that continue to influence decisions about the future, as demonstrated in Hein and Schubert [28], on the specific case of port cities. The existing urban practices can help or hinder future development; paradigms of growth or speed influence our decisions about which technologies to use. If one considers the science and technologies to monitor and plan land use, this junction between the past, the present, and the future is also relevant to support cultural perspectives; it highlights the absolute necessity of cooperation between different disciplines, to process sources with different kinds of uncertainties, possibly scarce when one goes back in time or simulated when one follows possible trajectories for the future. Such cooperation is mandatory to adopt critical perspectives on sources, even with a lack of ground truth, and to state explicit hypotheses, such as the radiative concentration pathways for climate scenarios.

This junction can be made more explicit and accessible to users by valorizing footprints from the past that have persisted until today and integrating them as historical data in city information infrastructures. These footprints vary. They can be intentional surveys, such as old maps and censuses, or other observations from the past, such as newspapers, postcards, and deeds, and they can also be tangible remains, such as monuments, soils, and landscapes, or intangible practices. At the Paris Urbanism Agency (APUR), the administration regularly visits archives, to identify precise past land use that can be responsible for the current soil pollution. Some hazardous waste can also be detected based on historical aerial imagery, such as unexploded bombs in Europe after World War II; craters are visible from the aerial campaign that was launched after the war, and have since been covered up. These craters are usually not visible in the recently produced images, because of land use since the war. Recent studies of the Venice lagoon flooding from Roman times until today, using memories, archival records, and underwater archaeological findings, show that the area constantly went through a 300-year cycle of low water, followed by the same time span for high water; it has been determined that we are now at the highest point of the high-water cycle [35].

This historical knowledge, grounded in observations and other data, should have informed the decision and design of the dam. Scientists have investigated such footprints from the past, to study the paradigms underlying our current cities' land use, or to evaluate a posteriori the impact of decisions. Hein and Schubert [28] examined port cities' path dependence and factors of resilience, using archival research. Long-standing institutional connections between port and city actors in Hamburg have led to a more balanced port-city development than in Rotterdam, where the port has traditionally taken precedence, or in London, where private actors left the historic city to pursue maritime activities. Alfasi et al. [36] compared the long-term land use plans established in 1982, with the actual land use extracted from aerial photos for the years 1980, 1990, 2000, and 2006, and found that the plans were inefficient. Benzerzour et al. [37] processed different plans, drawings, and meteorological observations from the past, to analyze the decisions made by the sanitation authorities of the city of Nantes in the 19th century, and the effect on the urban climate. They demonstrated a posteriori that their actions had a positive impact on the city's microclimate.

Current technologies are opportunities to make use of archives and other sources from the past into historical data on a large scale, while preserving the qualitative approaches developed at the local scales. To encourage culture awareness, it is crucial to support the evaluation of the specificities and distinctiveness of a situation, by comparing objects with no limit in space and in time. In the last decades, different mapping agencies in Europe, such as IGN France, the Dutch Kadaster, SwissTopo, and the Danish Geodata Agency, have digitized their archives of analogical maps and images, and have published them in national portals, to make them accessible to every citizen, either as single digitized sources (for images) or as a new integrated product, thanks to orthorectification and registration. These projects have been met with tremendous success among citizens and the media. In France, these materials are used in primary schools, to teach about important land use processes, such as the urban growth that has occurred with industrialization. In some locations, digital humanities projects investigate how to produce knowledge based on footprints from the past, with the assistance of computer science, adapting the big data paradigm to these sources from the past [10].

To conclude, grounding culture awareness of city information infrastructures in historical data, in a scalable way, is a pragmatic approach to associating a given land with its previous covers and usages, and can be used to identify related communities, thus adding more depth and other dimensions to the description of land use in information infrastructures. This improves the identification of the specificities and distinctiveness of a given land use, for example, the presence of industrial sites, by comparing not only the current status, including indicator values, but also additional dimensions, such as the sequence of indicator values and comparable situations in space and time. This is also a pragmatic approach to foster the integration of different technologies and disciplines related to land cover and land use. Processing and interpreting historical sources requires collaboration between people with the necessary expertise to interpret and process these data and identify related uncertainties.

### 3.2. Analysing Functions of Culture Aware Land Use Information Infrastructures: The LandUseWheel

### 3.2.1. Identifying High-Level Functionalities

This section analyses, in more detail, the functional requirements of information infrastructures, to develop the approach that is presented in 3.1. This analysis is based on the practice of digital humanities and geomaticians, as well as the expectations of practitioners.

Historians and scholars, in the digital humanities, have started designing historical data and information systems for a variety of tasks, including debating within scientific communities and producing new knowledge. They are currently focused on decisions regarding how to treat the data and on the information life cycle. The WaterWheel, proposed by Hein et al. [7], summarizes the findings from the Digital Humanities Group of the Delft University of Technology, who studied the design of an information system based on paper archives, to study the past and current dynamics of a port city. The authors analyzed the information life cycle and identified a wheel of successive steps that need to be taken by different groups as illustrated on Figure 1. The wheel refers to producing and processing raw data, and distributing the newly generated data, so that they can become a new source for other groups. The concept of the wheel speaks to the need for an agile methodology to facilitate mixed-method work in the digital humanities, where historic archival research is integrated with big data approaches, and where geospatial mapping of the past is linked to the design of the future. Such a circular approach has the potential to knit together the past, present, and future, as well as multiple disciplines.

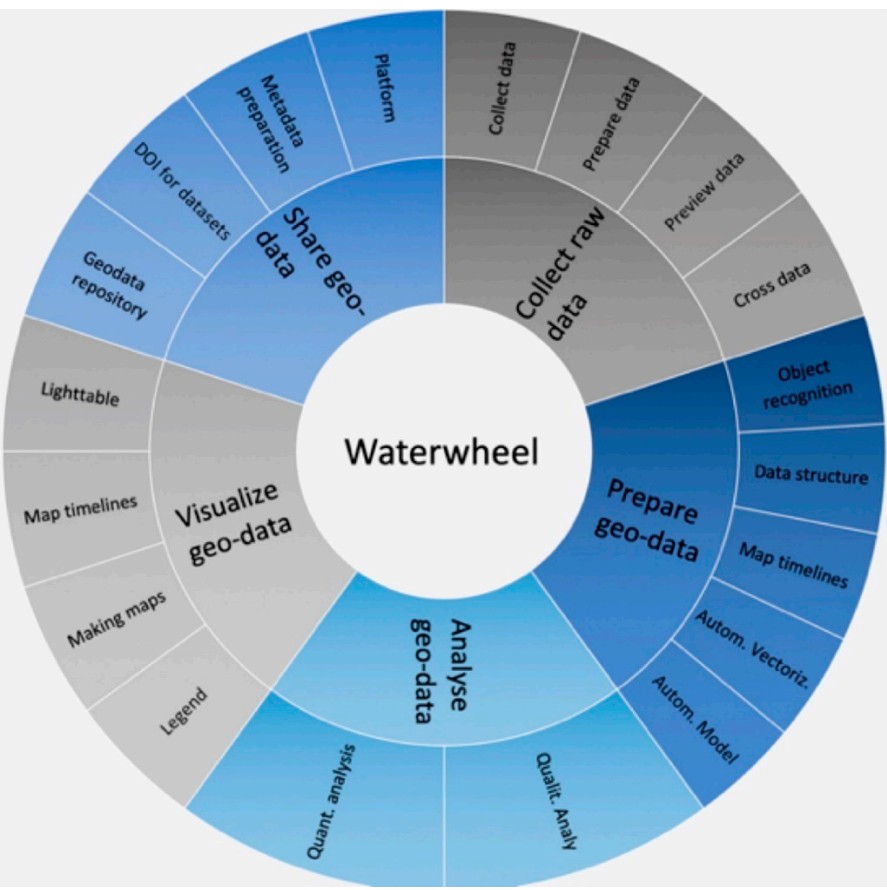

**Figure 1.** The WaterWheel symbolizes the life cycle of information and data in a digital humanities project [7].

Many scholars in the field of history are exploring ways to use new digital technologies from an interdisciplinary perspective. The International Commission for the History of Towns was funded to support the interconnection and transversal coordination among scholars studying the history of cities, who engage colleagues across cities and countries, as well as across disciplines. The commission investigated the following three different projects: national bibliographies on urban history, registers of early documents relevant to urban history, and national atlases of historic towns [38]. The latter project is of specific relevance to attempts to integrate historical data into information infrastructures. A national atlas of historic towns is a set of guidelines to create historical land cover and land use data of towns in a given country, considering the available sources. An alignment is sought between atlases, through the commission's Historical Towns Atlas Working Group. A core common portfolio was identified iteratively comprising a cartographic draft of the original maps and of a modern city map, at specific scales, to ensure comparability (1:2500, 1:25,000, 1:50,000), and including a pre-industrial cadastral map to depict the development of the town and a historical essay [38]. The group identifies several steps in the design of an atlas. An initial step is the selection of towns to be included in an atlas volume, with the goal to identify varied and characteristic towns.

In the domain of contemporary land cover and land use data production by cartographers, the classical steps involved the specification of the targeted land model, data acquisition, data processing and integration, data analysis, and data rendering. An important aspect soon became co-funding and achieving a consensus regarding a land model that could meet the need of several users who could pay for data acquisition and processing. In the 1990s, the need to decrease the costs related to data acquisition and maintenance, as well as to take advantage of the opportunities offered by distributed architectures, led to the design of the spatial data infrastructure. A key functionality of an SDI is the

referencing of data on a catalogue by a provider, and the discovery of relevant data on a catalogue by a user. Most recently, the field has evolved to include the design of search engines that are dedicated to datasets, and the adoption of search engine optimization by data providers [39,40]. Indeed, a key functionality has become the discovery of the most relevant dataset from a user who may not be familiar with the complex metadata model that has been adopted in the geographic information domain. Another way in which the cartographic production process has evolved involves the opening of the process to communities, at first local administration or fire fighters, for example, then the engagement of citizens in sending alerts to update the data and collaborative platforms for the very production of the data. In that context, a key function is the evaluation and documentation of data quality.

Below, we present a draft of a LandUseWheel for a smart city, which identifies five high-level functionalities illustrated on Figure 2. Three of them are derived from our analysis of the field of the smart city in Section 2.2, and are evolutions of the three traditional pillars of the smart city vision, namely, to make a more accurate representation of the city possible, to support decisions, and to enhance the interface with citizens. In these functionalities, we insist on the target of cultural awareness, which is as follows: the capacity to associate related communities, the capacity to evaluate distinctiveness, and the importance of supporting the engagement of stakeholders from different backgrounds. Another high-level functionality is knowledge production and dissemination, as well as territorial intelligence design, extending the evolution of the smart city discourse, as well as the WaterWheel and the FAIR principle. Finally, we consider the high-level functionality of producing historical data to be integrated in information infrastructures, as it is the core of our approach (Section 3.1). In this functionality, we adopt the WaterWheel principle and extend it, in order to emphasize the specification of the data model, to ensure scalability and to preserve the expressiveness of local specificities, in conformity with the Historic Towns Atlas approach and with the experience of land cover and land use data production by cartographers. The functionalities form a circle in an agile methodology that allows for different approaches to be pursued simultaneously, and then knitted together. From a scientific perspective, the production of knowledge is a trigger to ask more questions and produce more data to answer the questions, and, from a practice perspective, the creation of territorial intelligence is a way to secure funding for further acquisition, through private actors or taxes.

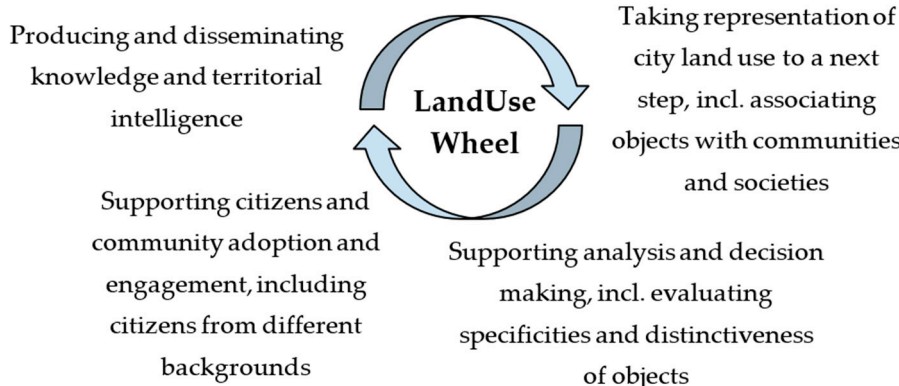

**Figure 2.** Our proposed LandUseWheel symbolizes the life cycle of information, data and knowledge related to land use. This circle only portrays the high-level functionalities that are detailed in Table 1.

**Table 1.** The LandUseWheel: functional requirements for culture-aware land use information infrastructures grounded in historical data.

| | |
|---|---|
| 1. Specifying and producing land use data and metadata for the past, present and future | • Identify relevant sources and historical data to study a given phenomenon<br>• Identify relevant models and referentials (in particular for partitioning and classifying)<br>• Digitize sources<br>• Preserving sources and securing acquisition (including future acquisition)<br>• Create metadata for the digitized source<br>• Specify and produce referentials<br>• Disaggregate and register a digitized source into historical data<br>• Create metadata for the historical data<br>• Publish historical data and metadata |
| 2. Taking representation of a city to a next step | • Create a model of the past state of a city's land use<br>• Interconnect past, present and future representations of a city's land use<br>• Create quantitative and qualitative junctions between multimedia content<br>• Browse and query historical data and documents through a spatial and temporal axis, through their content<br>• Connect different literacies<br>• Evaluate/understand source uncertainty |
| 3. Enhancing analysis, especially of specificities and distinctiveness, and decision capacities | • Compute comparable indicators<br>• Detecting changes, reconstructing sequences and paths<br>• Retrieve facts and situation from the past of Europe<br>• Retrieve/design collections (to learn or to get recommendations) |
| 4. Enhance citizen and community adoption and engagement | • Rendering a place for a specific user<br>• Query historical data through an intuitive interface<br>• Communicate and manage regulations related to cities in a spatial and temporal perspective |
| 5. Support knowledge production and dissemination, and territorial intelligence | • Document the scope of the produced knowledge<br>• Publish in a scientific community, publish a data paper<br>• Express knowledge in a shareable language<br>• Transpose a given proof of concept<br>• Identify new questions |

The remaining sections describe how these high-level functionalities were detailed in scientists' experience or in practitioners' expectations. These are practitioners who use information systems in their daily work in cities, and geomatics or IT specialists who assist them in using geographical data. They focus on advanced functions that could be expected, based on historical data. These practices and expectations are gathered from interviews conducted during the CSA Time Machine and from discussions during the Time Machine consortium meetings.

3.2.2. Specifying and Producing Land Use Data and Metadata for the Past, Present and Future

The creation of historical data can be triggered either by a specific research question or an operational need, in which case a sub-function is to identify the relevant sources, and to locate and access them. It may also be triggered by the will to preserve archives, by opening them to a wider audience online, in which case a sub-function is the identification of the relevant collection of sources to process by the source provider. For example, IGN France prioritized the processing of its archives based on the uniqueness of the source and the capacity to provide a homogeneous representation of the past. While in the domain of cultural heritage, digitization is applied only to the documents of high value or to prestigious monuments, and producing historical land use data may require processing more ordinary information.

The design of land use models requires the specification of an abstract conceptual model to identify relevant sources and process them. At a conceptual level, three paradigms exist to model lands. Object models explicitly represent phenomena that have shapes and boundaries, such as roads, buildings, or administrative units. Continuous field models are more adapted to phenomena that vary continuously in space, such as elevation. The third model, of "area-class maps"/"categorical coverages"/"chorochromatic maps"/"discrete fields"/"thematic regions", is adapted to represent the phenomena that are neither continuous fields nor entities with well-defined shapes, by giving these phenomena some spatial grounding, through a partition of space into areas (regular or not) and through a set of values describing the phenomena of interest. These values are usually indicators computed in a manner based on spatial statistics. This representation is useful to support decisions, because it can merge different thematic information into one representation. Areal representations are a dominant paradigm, in what are called land cover models, because of their capacity to support diachronic analysis if a stable partition of space and a stable classification are used, and hence they can be used to monitor the evolution of land cover. Despite space being a shared framework across disciplines, there is no universal land model with types of objects of interest, types of areas of interest, or types of fields of interests that are shared by everyone, even within what is called common-sense knowledge [41]. Hence, when creating a land model, the user must make ontological commitments to specify this information. During this stage, scalability can be anticipated by reusing already used models or by aligning an ad hoc model with existing ones.

Once raw sources are digitized, an important step that has been identified by scientists is to disaggregate the source, to represent the atomic pieces of information contained in it that will be processed to create the land cover and land use model, and to support automatic analysis, integration, and query. Reference systems are needed for this step—semantic, spatial and temporal—that will be used to reference the sources, possibly transforming them, and to support cross analysis and automated operations. Their selection must be guided by the land model at stake.

For these new disaggregated sources to be visible from open information infrastructure, it is important to associate them with metadata that provide hints about the semantics of the source, and the expected audience and message, possibly pointing out a given community that holds the required expertise to work on that source. More metadata can also be added in the remainder of the LandUseWheel, to support search and browsing.

Regarding the representation of future land use, an important expectation of practitioners is to have an overview of the possible local climate scenarios for their city. Currently, climate scenario models are available at a regional kilometer scale; more downscaling is required to represent distinct local micro-climates. Simulating the future of cities also requires evaluating the joint impact of physical climate change and of adaptation strategies [6]. A specific land cover and land use model has been adopted by the climate community, called local climate zones (LCZ), to characterize rural and urban tissue through classes that are homogeneous in their type of climate and that are representative of the real-world reality, at the resolution from 500 m to 1 km (LCZ0), and 50 to 100 m (LCZ1) [42]. The LCZ propose ten types of urban zones, as follows: compact or open high-rise buildings, compact or open mid-rise buildings, compact, open or sparse low-rise buildings, large low-rise buildings, and heavy industry and lightweight low-rise buildings.

One issue is securing the sources and the provision; this entails either protecting analogical archives that can be damaged, or ensuring the funding of acquisition in the future. This refers to the general domain of data governance.

### 3.2.3. Taking Representation of a City Land Use to a Next Step

Taking the representation of a city to the next step mainly consists in supporting the following:

- The junction between the past, present, and future, within information infrastructures;
- The creation of more-accurate models based on the integration of land cover and land use data available in the infrastructure;
- The association of an object of interest with relevant communities and societies.

The junction is expressed in practice as the capacity to search, access, browse, integrate, and query different datasets and documents through a spatial and temporal axis, whatever the provider and the type of dataset. It is expected that searching can be carried out across all the possible sources and data, based, at first, on spatial and temporal criteria. Access management can be either directly at the source or from an intermediate document and information provider. Browsing mechanisms should support switching from one view to another, and should propose a visual linkage mechanism to allow the cognitive task to switch from one to another. The relevance of this functionality to support historians' tasks was more experimental in ALEGORIA and in Archival City, developed for quantitative and qualitative purposes, respectively. Indeed, there are qualitative and quantitative junction projects. Qualitative junction is the capacity to precisely process the information contained in selected sources and is available on a limited site for which the solution is dedicated, as in Archival City for the Greater Paris or other sites. Quantitative junction is the capacity to embrace a large variety of sites in a robust way, without being disturbed by local differences, but sometimes with less discrimination than in qualitative approaches. As an example, ALEGORIA developed a quantitative solution covering the entire French territory in various periods, from the interwar period to the present day. Qualitative junction projects are dominant in the communities of historians, while quantitative junction projects tend to be led by computer scientists.

Together, ALEGORIA and Archival City have proposed a spatio-temporal web application that is dedicated to the co-exploitation of heterogeneous data spatialized in a common 3D environment, providing several paradigms for supporting their co-visualization and interactions within the 3D environment and across time, as illustrated in Figure 3. The relevance of this tool was demonstrated with several use cases involving historians and sociologists, with the common objective of better understanding the formation of the Parisian metropolis. The study focused on the evolution of the city of Nanterre (Paris area), which underwent many changes in the 1950s, and, in particular, it focused on shantytown areas. Through census as statistical data and aerial imagery as visual data, a group of historians and sociologists experimented with the relevance of the joint exploitation of those heterogeneous data within the spatio-temporal web application [9].

Another expectation from practitioners is the capacity to create a new dataset that represents a past state of a city component, such as a parcel or a building, to feed the user application. This requires the capacity to identify available data, and to compare them and their respective accuracies, and possibly to align and merge them. For example, in France, the reconstruction of accurate geometries of parcels sometimes required browsing textual documents, because cadastral maps were not always as accurate as today's digital cadaster and it was necessary to consult the valid source in the law. More globally, uncertainty is especially present when working with sources from the past, because they can refer to technologies and production contexts that different users are not familiar with. It is essential to integrate expertise from scientists who have studied such sources, and whose insight is necessary to adopt a critical perspective on them [43]. Creating a dataset that represents a past state of a city component may also require completing missing information. This is the case, for instance, when obtaining a virtual reality model of a past state when no true images are available, only drawings.

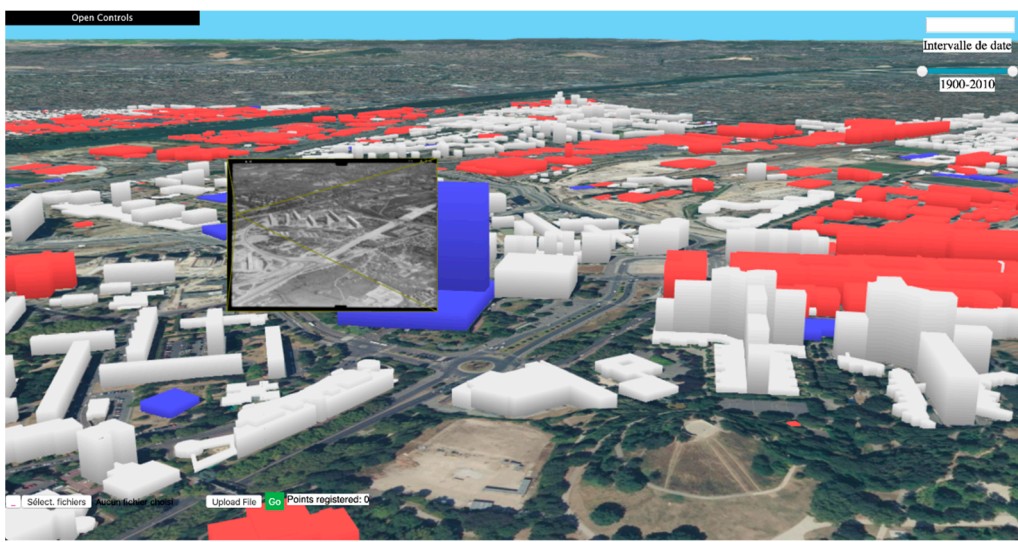

**Figure 3.** Free navigation in the 3D multimodal environment of Nanterre (vertical recent imagery, oblique historical imagery, statistical vector data and 3D buildings) (image from [9]).

### 3.2.4. Supporting Analysis and Decision

Analysis targets the understanding of land use dynamics and of tool efficiency, in a cross-level context and in the context of climate change.

Important expectations that are expressed by practitioners, regarding this high-level functionality of a culture-aware land information system, include the capacity to do the following:

- Put an object in perspective, by comparing it with other places, possibly distant in space and time;
- Detect unexpected behaviors and trends, and to understand dynamics;
- Obtain recommendations.

Practitioners need to compare lands and their evolution. They may want to compare land cover and land use indicators between different places or different time stamps, or the analysis of the evolution of specific spatial phenomena, such as urban sprawl, through its spatial or non-spatial characteristics. Expert users would expect the possibility to implement and run spatial statistics whenever a model exists, to describe the compared situations. However, in most cases, the comparison is rather is a semi-automated analysis, where the user relies on typologies to identify patterns, and then further investigates the patterns. Finally, detecting an unexpected behavior assumes that the system has a model of the expected behavior of the land. In another context, to improve the temporal resolution of the current land cover models, using new sources of data, the Austrian national land cover model now comprehends functions to describe some attributes that vary with time, in order to encode the expected behavior of pastures, for example, and to facilitate the detection of unexpected change.

Other expectations are more related to decision making, as follows:

- Design benchmarks, obtain recommendations regarding specific land planning decisions, based on what has already been experienced elsewhere;
- Simulate a given land use taken from another place, for example, white roofs or cycling paths.

They need to query the information infrastructure for facts and situations, as well as corresponding observations and data. The query could be expressed with similarity criteria, such as, for example, "cities with similar topography that underwent similar flood events" or "cities with similar meteorological conditions, but totally different architectures". Alternatively, the query can explicitly specify characteristics, such as "a city with two ports". The results must be clustered and ranked. The potential users are scientists who

study a phenomenon, and would like to sample European history for examples and data. Other potential users are decision makers looking for inspiration. In particular, low-tech traditional uses and practices can serve as inspiration for future sustainable developments. This requires skills in artificial intelligence, but also domain-specific skills, to make valuable comparisons, assess similarities and differences between contexts, and promote critical perspectives regarding the results.

### 3.2.5. Enhance Citizen Engagement

Citizen engagement can be enhanced through a user interface that renders the land's cultural specificities. The UbiSoft gaming company works hard on projects to render distant places. In their experience, this cannot rely on 3D models depicting only the landscape. Additional information must be simulated and added to the scene that makes up the place, such as people walking, noise, or light. Investigation is necessary to ensure that the added information is relevant to the depicted place, which improves the culture awareness of the representation.

It can also be enhanced by an interface that adapts to the user's specific background. Chatbots have been adopted by the Dutch Kadatser, to prototype user-friendly interfaces to query legal data [39]. The chatbots' conversational capacities can adapt to different wording and logics.

Finally, in the context of Europe, an expectation that was mentioned was the capacity to know what regulations come from Europe and apply them to a given place.

### 3.2.6. Knowledge Production and Dissemination

An especially challenging task is to document the scope of the derived knowledge and how it can be extended to other places [7], as the produced knowledge is aimed at being disseminated as longitudinal knowledge outside the original community. In the URCLIM project, the study of specific micro-climate phenomena, such as road icing or urban heat islands, is run on at least two different European cities in parallel, to improve the scalability of the results [6].

Another important aspect during knowledge production is the identification of new questions to be asked of the sources, which can possibly trigger new historical data production [28].

### 3.3. *Analyzing Technology Readiness of Multiple Modality and Metadata for Culture Awareness*

This section identifies three key technologies to serve the above functions on information infrastructures, and to analyze their technology readiness.

### 3.3.1. Structuring Multimodal Content

Multimodality, or the capacity to search, browse, and integrate heterogeneous media, is of crucial importance to support the following different functionalities that are analyzed in Section 3.2: enhancing the analysis and the capacity to compare situations, supporting the disaggregation and annotation—e.g., authentication—of digitized sources, enhancing the querying of data.

The content that is managed by the information infrastructures can be sources from the past, digitized in the form of poorly structured information, or they can be disaggregated content, possibly structured or semi-structured data after reconstruction, and also digital-born sources. Multimodality exploits the image content characteristics, or textual documents or structured data, in particular land cover and land use data. It requires first building links between these representations and eventually aligning them.

Spatialization technologies are useful for building these links. Many of those contents can be associated, directly or indirectly, with spatial information, from a simple street name to a position and orientation in 3D space. This information may be of great importance, because it provides a common framework for their joint analysis and exploration, and, in particular, it may facilitate their exploration through the time axis. Classically, in the

SSH and DH communities, solutions allowing the joint exploitation of these spatialized data relies on their manual registration on a 2D map with a GIS (geographic information system) most of the time, layer by layer.

Content-based image retrieval (CBIR) is a domain that bridges image analysis, learning, computer vision, and databases. Typically, the images of a collection are described and indexed through the analysis of their content, and the descriptors provided are organized to compare and retrieve images, or part of images, efficiently in a dataset, at a large scale. This querying paradigm is complementary to classical textual indexes, and the literature on this research domain is extensive [44,45]. With the advent of deep learning, new proposals regularly emerge, with the main objective of treating collections in difficult conditions (time difference, change of season, sensor, viewpoint, day/night, etc.) [46]. Sometimes, the images can be associated with other multimodal contents, such as semantic annotations (free text or structured keywords), or even geometry (e.g., LiDAR, RGBD sensors, maps), when considering landscape retrieval. The literature on multimodal search engines mixing text and images is extensive [44,47,48]. When it comes to image content in remote sensing and landscape search, multimodal retrieval is generally based on the joint exploitation of data from different sensors [49] or involving vector maps [50]. The most common strategy, especially when mixing text and images, is to transform the "cross-modal search" problem into an "intramodal search" problem, and to learn, from the data, a common latent space, in which data from different modalities can be projected using linear projection mappings [51].

The applications of CBIR are various in all areas where poorly structured large sets of images must be manipulated. Concerning image contents related to cities, the GLAM holds many iconographic archives representing cities, as well as mapping agencies and private companies that regularly map cities and their evolution. Here, CBIR is at the core of the management of those image datasets, which can be quite voluminous and poorly organized, in terms of structure, annotations, and interlinking. CBIR may help in browsing those contents independently of the potential existing annotations, within a collection and across collections [52], and, more generally, in providing interconnections between contents, in order to improve global consistency and propagate relevant information. Nowadays, one timely research topic concerns the localization of visual contents, which can be solved by content-based image retrieval in a collection of georeferenced images. With the development of AI, more-advanced techniques exist to determine a localization. For example, with images in computer vision, and photogrammetry and robotics domains, solutions exist for estimating the pose (position and orientation in the scene) of the camera at the origin of the image; the most recent approaches exploit all the available information (3D models, 3D point clouds, etc.) [46] in a fully automatic or semi-automatic way [9].

It is necessary to articulate the following two types of approaches: quantitative solutions that process large volumes of information on a large scope, and qualitative approaches that are dedicated to the specificities of a given site. Articulation can be accomplished either at the level of a given site, by switching from a quantitative solution to a qualitative one, or it can be accomplished by comparing different qualitative solutions, in order to identify commonalities, e.g., in the format or with CBIR tools.

### 3.3.2. Land Data Models and Knowledge Graph

A key technology for the LandUseWheel is the capacity to achieve scalable and expressive land data models. A recurring issue in land data models is the lack of ability to partition space, in order to generate portions that will be homogeneous in terms of land cover. Homogeneity can be considered at the level of one portion of space (one area). State-of-the-art models are proposed, to account more for heterogeneities that can occur in 3D. Devos and Milenov [53] apply, to land cover models, the concept of tegon as a "horizontally homogeneous, physical spatial object with a notable spatial dimension and a specific life cycle, characterized by the presence of a substrate and possibly one or more vertical biotic or abiotic strata". The homogeneity of the model refers to the fact that its

accuracy is comparable throughout the model. Authors also investigate smarter zoning and land use concepts, to better adapt to the complexity of our cities [54,55].

A field that is especially relevant to the LandUseWheel is the Semantic Web. The basic idea behind the Semantic Web is to use the web to express shareable pieces of information about reality, through three main principles. First, one should use dereferenceable identifiers for depicted resources, and the web resource corresponding to the identifier should identify the resource with sufficient clarity, such as "https://en.wikipedia.org/wiki/Venetian_Lagoon" or https://www.wikidata.org/wiki/Q76925. Secondly, one should express statements in the form of graphs. Thirdly, one should express links between their statements and other statements, to improve the navigability across statements, and to improve the design of transversal query solutions. These models are particularly useful for depicting a complex world, through a so-called knowledge graph, e.g., "Mona Lisa is a famous painting by Leonard Da Vinci who was born in Vinci". In countries such as the US, Ireland, the Netherlands, or Spain, national knowledge graphs are being developed to encode connections between entities in the real world that are represented in different authoritative databases, to achieve a uniform representation of a given space that can be queried in a seamless way. This technology is a powerful way to interconnect data across different domains, formats, and sources [56].

### 3.3.3. Spatial Data Infrastructures and Metadata

Spatial data infrastructures (SDI) target the availability and reuse of spatial data across different providers and users. They are not only data and technologies, but also policies, institutional arrangements, and people [57]. In the early 2000s, the advances of information infrastructure technologies and the rise of geoportals, including the European Environment Agency portal to distribute CORINE land cover data and national geoportals, led the European Parliament to establish an information infrastructure for spatial information in Europe (INSPIRE), in order to ensure consistency and trust at all levels, from the region to Europe [31,58]. The launch of INSPIRE was, in part, motivated by the fear of seeing de facto private information infrastructures become the reference to establish and monitor European environmental policies [58]. The current implementation of the INSPIRE directive is too complex and under-used. Kotsev et al. [59] recommend that existing "SDIs [should] dissolve in emerging data spaces defined by the European Commission" and that "all actors should become first-class citizens who participate in the co-design and co-creation of technological solutions". At the scale of Europe, an important result of INSPIRE is the identification of common data schema that are aligned with national schemas for public data in all the domains related to the environment, identified by the following directive: all geographical data (e.g., geodetic coordinate systems, imagery, topographic data), but also statistical and meteorological data. It is also the production and provision of metadata about the corresponding data. These alignments of national models and these metadata are a wealth yet to be exploited for the required functionalities, to catalogue, compare sources, and assess uncertainties. Yet, the main flaws of geographic metadata are their heterogeneities, as well as the lack of means to easily produce them. Ref. [39] argue that the achievement of successful SDI requires an improved management of semantic heterogeneities and a better exploitation of metadata, in particular INSPIRE metadata. They apply the VGI paradigm to metadata and propose the collaborative design of an open knowledge graph of digital assets in Europe, which will contain metadata about assets; alignments between these metadata, to ease their homogenization; and alignments, at the metadata level, between assets. Such a collaborative platform, to produce and share metadata as linked data, addresses the following life cycle of metadata: production, curation, and usage.

Besides metadata, other models are needed to support the comparison of two land models. The EAGLE model (EIONET Action Group on Land Monitoring in Europe) has been defined to annotate the LC LU classification system, to facilitate comparison and integration. It is based on descriptors (object-oriented) and is scale-independent. The

principles developed by the EAGLE group have been adopted by the ISO 19142-2 standard called LCML (land cover meta language). This LCML is recommended by the INSPIRE data specifications for theme LC. The comparison of abstract models can also rely on high-level ontologies, such as DOLCE (descriptive ontology for linguistic and cognitive engineering) and BFO (basic formal ontology), which has been extended and adapted to the geographical data [60,61]. In DOLCE, the temporal characteristics of phenomena are essential for distinguishing between high-level categories, such as "enduring" (or continuing), which includes entities that have an identity that endures over time, such as a city or a person, and "occurring" (or lasting), which includes events or processes.

## 4. Discussion

The vision of a culture-aware information infrastructure that is capable of querying the diversity of the land use experiences of Europe, as an open knowledge book, and comparing land across space and time, is appealing. This paper proposes a formalization of that vision, with contributions that come from the different disciplines of the authors, and it develops a specific target of culture awareness and a LandUseWheel framework that can help information infrastructure move towards this target. A detailed analysis of the LandUseWheel highlights the opportunities that are offered by the current technologies that make use of multimodality, land models, knowledge graphs, spatial data infrastructures, and metadata. Of particular relevance is the identification of open, interoperable and scalable solutions that will complement the existing systems and not replace them. The priorities are the design of shared identifiers' policies among communities, for the objects of interest in the smart city land use, and the creation and sharing of alignments of land use models and of metadata for CBIR. More integration between data and metadata is necessary to facilitate access to metadata and their usage. Many issues remain open. Including data from the past in smart cities projects can be viewed as extending digital twins' capacities to embrace past states, and potentially to help understand shared paradigms that have driven decision making over time. Including long-term dynamics, as well as dynamics that cannot be simulated by physical models, is a big challenge for simulation. The dominant paradigm in big data is to use real-time sensors to improve the currency of the representation. These data usually have similar structures and are well suited to machine learning algorithms. However, data are less structured, less dense, and scarcer when one goes back in time, and may face evolutive representations for which training data are not adapted.

In a distributed and open context, authentication and trust are two important concerns. Our capacity to assess the veracity of a source varies depending on the technology, but also on the conditions of the production. When it comes to tangible sources, as is often the case for historical data, specific authentication methods have been developed. Practices exist in science and in regulation to identify and quote data, and to refer to specific authorities. However, these practices are too-often siloed. We lack curation support in the field of smart cities and need a unifying authentication infrastructure for all digital objects.

Retrieval capacities are needed to search and rank results. It is crucial that the retrieval models are transparent enough for users to understand what criteria and biases have been adopted, to search the past and to rank the results. The same applies to learning samples.

Securing the provision of data also remains an issue. Whereas scanners and related programs exist, innovation is required to propose solutions at an affordable cost for cities. It is also necessary to evolve the core mission of holding institutes that have rights over the records to share and open them.

## 5. Conclusions

While culture often is acknowledged as an important feature of societies, it is rarely viewed as key to sustainable development. This paper expands upon the UNESCO definition of culture, to search for a solution to the challenges faced by the smart city domain. Culture awareness can contribute to meeting these challenges, by improving dialogue and collaboration among all stakeholders, including those in the sciences and in practice. A

primary contribution of this paper is to formalize a target of culture awareness for city information infrastructure.

A second contribution of the paper is the LandUseWheel, which is a conceptual framework to progress towards the target of culture awareness, through the valorization of historical data. To facilitate land use understanding and planning, we need better and more complex data on the past, and we need to analyze it in a coherent, standardized way. The purpose is to reason and debate, not to imitate the past, but to learn from it, to be more inclusive and also more aware of our wealth in Europe, and to be creative and resilient. Our heritage tells us that our societies are capable of transformation and resilience, which may help stakeholders to accept the transformations that are required of our societies [62]. Increased digitization of quantitative and qualitative data of the past and novel computer science technology (crowdsourcing, AI), allow us to put historic data sets into conversation and to develop long-term analyses. The current evolution of information infrastructure could be cultivated to accelerate the transition towards such smart sustainable cities. This is what we call culture awareness of city information infrastructures.

Integrating historical data is also an opportunity to foster collaboration among communities and to help the current projects evolve, even when handling contemporary data. The expression "historical data" refers, in these examples, to the past, but also applies to current data that will become historical in the future and must be managed as such, so that successive generations can shape their futures.

Future work will concentrate on supporting the design, by relevant stakeholders, of shared land model specifications for past and future land data, including identifier policies, and of alignments between these models. As the overall scope of such culture-aware information infrastructure is too wide to let a consolidated community emerge, we need catalysts, with a smaller perimeter, where synergy is fostered by historical data. Two categories of catalysts are already envisioned, based on the CSA Time Machine project experience. The first is that of thematic catalysts that aim at building communities around land use themes for which history makes sense, as well as culture, such as public space. The second category is that of specific sites where catalysts could foster agreements between institutions who hold the data on a given site.

**Author Contributions:** Conceptualization, investigation, writing—original draft, writing—review and editing, Bénédicte Bucher, Carola Hein, Dorit Raines and Valérie Gouet Brunet; funding acquisition: Bénédicte Bucher and Valérie Gouet Brunet. All authors have read and agreed to the published version of the manuscript.

**Funding:** This research was partly funded by the project ERA4CS URCLIM funded by EC, by the ALEGORIA project (grant ANR-17-CE38-0014-01) funded by ANR (the French National Research Agency) and the Archival City Tremplin project funded by I-SITE FUTURE.

**Acknowledgments:** We gratefully acknowledge participants to the CSA Time Machine for fruitful exchanges and discussions during this ambitious project.

**Conflicts of Interest:** The authors declare no conflict of interest.

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
