# Peer review of "Towards Culture-Aware Smart and Sustainable Cities: Integrating Historical Sources in Spatial Information Infrastructures"

_ijgi, doi:10.3390/ijgi10090588_

Round 1

Reviewer 1 Report

If I understand correctly, the purpose of this paper is to make the case that cultural data, specifically historical land-use data, should be an integral part of Smart & Sustainable Cities projects, and you introduce the LandUseWheel as a conceptual framework for how to integrate it.

This paper is very well-written, eloquent in several places (esp. the conclusion), and very well-sourced. Rhetorically, most of the major claims of the paper are well-justified. The only challenge I have is in finding sufficient novelty to be worth publishing as something more than a literature review essay. It is not an original research paper. If that is acceptable, then it is publishable. I'll let the editors make that determination, so on to more specifics.

Brief summary of Smart Cities initiatives: great

Justifying that Culture is relevant to Smart Cities: great

Justifying that Land Use History is relevant to Smart Cities: good

However, I'm left unconvinced that Land Use History is directly tied to Culture, at least in the example projects you list, and wondering if all the part about Culture is even necessary. Why not just leave it out, since you already make the case that Land Use data would be useful, without all the business about UNESCO. I realize that the Venice example is supposed to make that connection, and it is a compelling narrative, but I don't see how the kind of historical land use GIS data you would have would be directly applicable to the issue (I can guess, but you don't tell me). The fact that it is historical does not by itself make something cultural.

The primary claimed innovation in the paper is the LandUseWheel framework, but I had difficulty seeing how it (or the previous Waterwheel) is any different than commonplace information infrastructure processes. Yes, I have not seen it presented in exactly these terms, but I have seen the same ideas in other forms, and GIS professionals and others around the world are doing these kinds of steps every day; is the conceptualization itself that novel? If so, then you need more literature review of past frameworks for information infrastructure processes to demonstrate that you have discovered something new.

The nitty-gritty (There isn't much):

Line 127: this is a very long paragraph. Perhaps split at "In 2018," or set the two long definitions as block quotes.

Line 483: You have three kinds of conceptual models: objects ("object-oriented" is too programming-laden), continuous fields, and neither of the above. That is unsatisfying, especially when there is an existing third category that clearly includes land use. The ontology of "area-class maps"/"categorical coverages"/"chorochromatic maps"/"discrete fields"/"thematic regions" is not as mature in the literature as the other two (evidenced by having five or more synonymous names), but it is out there and should be used more.

Line 566: I am doing some work on the ontology of landscapes, and I love the idea of "urban and rural tissue." I'm going to investigate this further. You also turned me onto this tegon concept.

One challenge of the approach of this paper, basically a literature review fit into a new conceptual model, is that you have to introduce a lot of projects and trends without enough space to really introduce them thoroughly. One example line 707. You should be introducing this in the context of the Semantic Web and Open Linked Data (and Knowledge Graph) initiatives, rather then triples/RDF (since the former are the infrastructure, the latter just the technical specification). However, anyone who isn't familiar with this field would leave this single paragraph confused, and adequately covering it would take a page. Could it be explained more simply?

I really look forward to seeing some of the future work ideas in the conclusions come to fruition.

Author Response

Thank you for reviewing our paper, please see the cover letter for a brief summary of the revisions and below more detailed answers to your review.

As you suggested we had a dedicated edition of the English language.

As you point out, the justification that Land Use management has to do with Culture has been improved. In particular we insisted on the adoption of a broad definition of Culture and we added more examples like that from Venice. We also made more clear we hope the connection between historical data and culture. 

We added more litterature review when introducing the LandUseWheel, in particular from the domain of cartography which was totally missing.

We adopted your terminology for the third category of model that indeed embrace several synonymous names.

We introduced the notion of Semantic Web and Knowledge Graph as you suggested, trying to keep it simple.

Thank you for your nice comments and for giving us the opportunity to improve our paper.

Reviewer 2 Report

General comments

The aim of the paper should be underlined more precisely. We can read that the paper "focuses on land use" (55), but there is no direct statement of what the Authors want to achieve with it or what are their hypotheses. The same applies to the discussion/conclusion where unanswered questions and visions for future works are underlined, but there is no explicit statement of the Authors contribution to the field. In general, the article collects many statements from the literature (rather superficially) and proposes a "LandUsewheel" which is an interesting concept. It should be definitely "put to the front" of the paper and related to other methodologies of spatial data gathering (GIS, Knowledge representation, cartography, DH). 

Detailed comments

"land use" notion should be defined in the article as it can be differently understood 

The Authors should describe what they mean by "digitising" (498). Is it scanning or vectorising? One could argue that scanning is not so expensive and scans of maps can be fairly easily shared via the Internet, while vectorisation of maps' content indeed requires more effort. 

The Authors present two projects: ALEGRIA and Archival City. I think they should be presented earlier in the text, but this is secondary. The primary is that the Authors missed one of the most important projects i.e. "Historic Town Atlas" (HTA) which presents the history of a given city in cartographic form, also, for several years, digital. I think HTA could be a good reference for a Smart and Sustainable City.

Author Response

Dear reviewer, Thank you very much for reviewing our paper. Please find attached a cover letter that summarize the revisions proposed to the paper and below more detailed answers to your review.

We improved the research design by being more explicit about our hypothesis, and positioning the LandUseWheel within the litterature. We described more precisely the context of the work, the adopted method and summarized the first results.

The introduction as been thoroughly revised to underline more precisely the aim of the paper and the two main hypothesis, and in the conclusion we explicitely summarized contributions.

We insisted more on the LandUseWheel in the introduction and added some litterature review in the text to insist on the novelty of it.

We explain which definition of land use we refer to and also give many examples.

Wrt digitising, we modified the paper to keep that word to refer to scanning and used the word processing otherwise. Yet, as you mention, digitising itself is not necessarily a costly process. To avoid adding still more words to the paper we chosed to remove the statement.

Thank you very much for pointing out the HTA initiative. We started looking at this project and going through publications. We hope it is correctly referenced in the new version of the paper.

Reviewer 3 Report

First of all, I would like to congratulate the authors for their work, I really liked the article.
I liked the article very much. However, there are some considerations that I would like you to
I would like you to take into account:

1- It is quite a long article with many topics. You should provide
the most relevant references in sections such as the discussion, the context of the research or the
context of the research or the Application to land use.
2- You can reread the article and avoid repetitions of text. "As mentioned
mentioned above..." if it has already been mentioned, there is no need to repeat it.
3- You should also avoid speaking in the first person plural. Use an
impersonal style.
4- On the other hand, I don't understand why you head "Materials and method" in
section 2 and talks about methodology in section 3.2.1.
5- In order to clarify your objectives from the beginning, you should clearly state your
hypotheses and research questions in the last paragraphs of the introduction.
paragraphs of the introduction.
6- You should also use the referencing style suggested by the journal.
journal. I attach a link.
https://www.mdpi.com/journal/ijgi/instructions#references

Any comments you wish to refute are welcome, it is not necessary to make all the changes if you think it is
to make all the changes if you think it is correctly explained.
Thank you very much for your contribution.

Author Response

Dear reviewer,

Thank you very much for reviewing our paper and for your congratulations, we are glad that you liked it.

We have improved the introduction and conclusion as requested.

The most relevant references have been introduced in the context or the research and application to land use.

An editor dedicated to english style has corrected the repetitions. Yet we did not succeed in adopting an impersonal style everywhere. We hope the current version is better though.

Indeed the section 2 title was here the one from the MDPI template but we have modified it to be more consistent with section 3.

We explicitely stated our research hypothesis in the introduction and the purpose of our work.

We revised the references style. Yet, within the text, we kept the same style and will put the number later if the paper does not need anymore revision because as we use word it is complex to handle bibliography.

Many thank you for the review and we hope we met your request.

Round 2

Reviewer 2 Report

I have no further comments. The authors have amended the article. 

Reviewer 3 Report

Thank you very much for your corrections. I see that you have made an effort to improve the document. However, the references are still not adapted to the journal. I will suggest minor changes so that they can be adapted and the article can be published. 
Thank you very much for your work.